# Histological Evidences of Autograft of Dentin/Cementum Granules into Unhealed Socket at 5 Months after Tooth Extraction for Implant Placement

**DOI:** 10.3390/jfb13020066

**Published:** 2022-05-25

**Authors:** Masaru Murata, Md Arafat Kabir, Yukito Hirose, Morio Ochi, Naoto Okubo, Toshiyuki Akazawa, Haruhiko Kashiwazaki

**Affiliations:** 1Division of Regenerative Medicine, School of Dentistry, Health Sciences University of Hokkaido, Tobetsu 061-0293, Japan; 2Division of Fixed Prosthodontics and Oral Implantology, School of Dentistry, Health Sciences University of Hokkaido, Tobetsu 061-0293, Japan; yukito@hoku-iryo-u.ac.jp (Y.H.); ochident@hoku-iryo-u.ac.jp (M.O.); 3Laboratory of Molecular and Cellular Medicine, Faculty of Pharmaceutical Sciences, Hokkaido University, Sapporo 060-0812, Japan; nao10okb@pharm.hokudai.ac.jp; 4Industrial Technology and Environment Research Development, Hokkaido Research Organization, Sapporo 060-0819, Japan; akazawa-toshiyuki@hro.or.jp; 5Division of Maxillofacial Diagnostic and Surgical Sciences, Faculty of Dental Science, Kyushu University, Fukuoka 812-8582, Japan; kashi@dent.kyushu-u.ac.jp

**Keywords:** human, dentin, cementum, graft, bone regeneration, DDM, unhealed socket

## Abstract

The aim of this clinical case study was to observe biopsy tissues at 5 months after an autograft of a partially demineralized dentin/cementum matrix (pDDM) into a tooth-extracted socket exhibiting healing failure. A 66-year-old female presented with healing failure in the cavity for 2 months after the extraction (#36). Initial X-ray photos showed a clear remainder of lamina dura (#36), a residual root (#37), and a horizontal impaction (#38). The vital tooth (#38) was selected for pDDM. The third molar crushed by electric mill was decalcified in 1.0 L of 2.0% HNO_3_ for 20 min and rinsed in cold distilled water. The pDDM granules (size: 0.5–2.0 mm) were grafted immediately into the treated socket. X-ray views just after pDDM graft showed radio-opaque granules. At 5 months after pDDM graft, the surface of regenerated bone was harmonized with the mandibular line, and bone-like radio-opacity was found in the graft region. The biopsy tissue (diameter: 3.0 mm) at 5 months after pDDM graft showed that mature bone was interconnected with the remaining pDDM. The novel histological evidence highlighted that newly formed bone was connected directly with both dentin-area and cementum-area matrix of pDDM. We concluded that pDDM contributed to the regeneration of bone in the unhealed socket, and this regeneration prepared the socket for implant placement. Autogenous pDDM could be immediately recycled as an innovative biomaterial for local bone regeneration.

## 1. Introduction

Extracted teeth are usually discarded as potentially infectious medical waste. However, the dental community is now considering human primary and secondary teeth as a useful material resource for patients and their families [1]. The first clinical case of human dentin autograft was reported in 2003 [2], while human bone autograft dates back to 1820 in Italy. There was a very long time-lag between the autografts of dentin and bone. Demineralized dentin/cementum matrix (DDM) is a decellularized matrix prepared through demineralization (<pH 1.0), washing by saline or distilled water, and/or freeze-drying [3]. Recently, many autograft cases were reported from the world [4,5,6,7,8,9]. Additionally, familial grafts between a parent and a child were successfully achieved, and the first successful procedure occurred in South Korea [10]. In this century, the tooth-derived materials have become a realistic alternative to bone grafting [7,11].

Highly calcified tissues such as fresh cortical bone and dentin do not occur earlier in osteoinduction and bone formation than DDM and demineralized bone matrix (DBM) [12,13,14,15]. In 2016, 70% of demineralized human DDM granules (size: 1.0 mm) had a better performance in bone formation than completely DDM (cDDM) and non-demineralized dentin in rat calvarial bone defects [16]. In 2021, cortical bone plate treated supersonically with acidic electrolyzed water (pH 2.3), so-called partial DBM (pDBM), induced bone at 2 weeks, while fresh cortical bone plate did not induce bone even at 8 weeks [17]. In light of the articles [13,14,15,16,17], we believed that the apatite crystals of both calcified dentin and cortical bone should inhibit a release of bone morphogenetic proteins (BMPs) from matrices. Clinically, the immediate DDM autograft system needs speed, compared to the delayed autograft. Our team, therefore, developed the immediate autograft system of partially demineralized dentin/cementum matrix (pDDM) by using a new electric mill [3,18].

The aim of this case study was to observe the biopsy tissues obtained from the dental implant-placed region at 5 months after pDDM autograft into the socket that had exhibited healing failure (#36) for 2 months. The histological evidence highlighted that new bone was connected directly with not only dentin-area matrix, but also cementum-area matrix of pDDM.

## 2. Materials and Methods

### 2.1. Clinical Case

A 66-year-old female presented with healing failure in the extracted cavity (#36) for 2 months after the extraction in a private dental clinic. The patient was introduced to our dental clinic in 2009. Clinical diagnosis revealed healing failure in the socket. Initial X-ray photos showed a clear remaining of lamina dura (#36), a residual root (#37), and a horizontal impaction (#38) (Figure 1). Her medical history was unremarkable.

### 2.2. Surgical Procedure 1 and Preparation of pDDM

First, both #37 and #38 were extracted under local anesthesia. The impacted vital-tooth (#38) was selected for pDDM, while #37 was a non-vital. The third molar (#38) was crushed with saline ice blocks in zirconium (ZrO_2_) vessel at 12,000 rpm for 1 min by our newly developed electric mill (OSTEO-MILL™, WiSM Mutoh Co., Ltd., Tokyo, Japan) [15,16]. The crushed tooth granules were immediately decalcified in 1.0 L of 2.0% HNO_3_ solution for 20 min. The pDDM granules including cementum (size: 0.5–2.0 mm) were extensively rinsed in cold distilled water (D.W) (Figure 2).

After the removing of epithelium and the cleaning of the socket by curettage, perforations into the dense bone in socket were performed by a curbide bur (Figure 3A), and the pDDM granules were grafted immediately into the treated socket (Figure 3B). The flap after relaxation incision was sutured with nylon threads (Figure 3C,D). The patient only took antibiotics for 2 days.

### 2.3. Surgical Procedure 2 and Tissue Biopsy

At 5 months after surgery 1, tissue biopsy (hole: 3.0 mm) was carried out from the central area of a fixture placement for the tissue observation under local anesthesia (Figure 4A–C). Next, a titanium-fixture (HAp coating type: diameter, 4.2 mm; length, 10 mm; POI system, KYOCERA Co., Ltd., Kyoto, Japan) was implanted into the regenerated bone (Figure 4D), and the flap was repositioned. The patient took antibiotics for 3 days. There were no troubles until the finishing of whole treatments.

### 2.4. Radiographic Evaluation

Initial X-ray photos including CT were taken in 2009 before pDDM graft. Next, X-ray photos were taken just after the pDDM graft (Figure 3D) (Surgical Procedure 1), and before and after the fixture placement (Surgical Procedure 2).

### 2.5. Tissue Preparation

Biopsy tissue was fixed with 10% neutral buffered formalin, decalcified for 7 days with 10% formic acid, and embedded in paraffin. Sections (with a thickness of 4 μm) were stained with hematoxylin and eosin (HE).

## 3. Results

### 3.1. Gross View and Radiographic Evaluation

Before the pDDM graft (Surgery 1), the lamina dura (#36) remained clear. (Figure 1A,C). Just after the pDDM graft, pDDM were seen like radio-opaque granules in the treated socket (Figure 3D). At 5 months (Surgery 2), a smooth surface line was seen on the graft site after the opening of mucoperiosteal flap (Figure 4A). The grafted pDDM was harmonized with the mandible, and a bone-like radio-opacity was found in the graft region (Figure 4B). A titanium-fixture was placed properly after the biopsy (Figure 4C,D). The final crown was set (Figure 5A,B) and functionality was well maintained. During the whole follow-up period, a complication, such as marginal bone loss, did not occur (Figure 5C).

### 3.2. Histological Findings of Biopsy Tissue at 5 Months after pDDM Autograft

The biopsy tissue showed that mature bone was interconnected with the remained pDDM granules (Figure 6A,B). pDDM included a small patch of cementum. The regenerated bone was connected directly with dentin- and cementum-area matrix of pDDM residues (Figure 6C,D). The dentin-derived areas revealed dentinal tube spaces and acellur matrix, while the cementum-derived area showed Sharpey’s fibers structure (Figure 6D). The boundary line between the dentin- and cementum-area matrices was seen clearly (Figure 6C,D).

## 4. Discussion

The biopsy tissue at 5 months after pDDM graft showed that the pDDM were received by the host and were harmonized with bone. Very interestingly, the histological evidence highlighted that newly formed bone was connected directly with dentin- and cementum-area matrices of the pDDM. This is novel evidence related to the direct bonding of new bone and demineralized cementum in clinical cases. We found that pDDM facilitated its adaption of the grafted site and was slowly absorbed as new bone began to form. As pDDM granules contain non-demineralized core (calcified matrix), the absorption of pDDM should take much more time than that of cDDM.

Dentin and bone are mineralized tissues and almost similar in chemical components. As bone and dentin consist of fluid (10%), collagen (20%), and apatite (70%) in weight volume [2,11], our attention for biomaterials focus on natural components such as collagen and apatite materials [19]. Both freeze-dried cDDM and cDBM are predominately composed of type I collagen (95%) and the remaining of non-collagenous proteins that contain a small amount of growth factors [20]. Both mature and immature types of bone morphogenetic protein-2 (BMP-2) were detected in human dentin and dental pulps by western-blotting [21]. Even after the demineralization of dentin and bone, active types of BMPs bind to acid-insoluble and collagen-rich matrices [11,22]. Interestingly, the strong acid treatment for bone- and dentin-derived materials increased their osteoinductivity and decreased their antigenicity [23]. Reduction and alkylation of BMP molecules resulted in the total loss of biological activity [24]. Additionally, after the processing system with 2.0% HNO_3_-demineralization as shown in Figure 2, bacteria free of pDDM granules were confirmed in the blood-agar medium [25]. From clinical points of view, sterilization by strong acid and safety are very important evidences for the processing procedures of hard tissue-derived graft materials [23,24,25]. After the immediate autograft of pDDM, thw patient was successfully restored to health with a dental implant. In the near future, ultrasound imaging will support histological findings and/or radiographic appearances as a non-invasive diagnostic tool [26].

## 5. Conclusions

The biopsy tissue at 5 months after immediate autograft of pDDM demonstrated novel histological evidences that new bone was connected directly with dentin- and cementum-area matrix of pDDM. A vital tooth-derived pDDM granules contributed to regenerate bone in an unhealed socket in a 66-year-old woman. After the pDDM autograft, the patient was successfully restored with a dental implant. Autogenous pDDM could be immediately recycled in our system as an innovative biomaterial for local bone engineering.

## Figures and Tables

**Figure 1 jfb-13-00066-f001:**
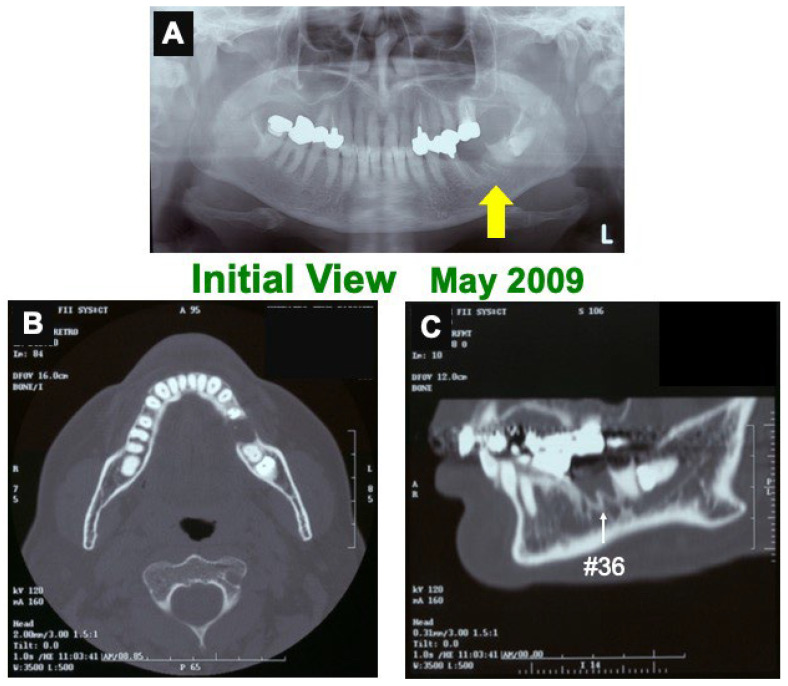
Initial views of panoramic X-ray photo (**A**) and CT (**B**,**C**). (**A**) Arrow indicating clear remaining of lamina dura (#36 socket). (**B**) Horizontal-axis view showing frame of mandible and non-bony socket (#36). (**C**) Sagittal-axis view showing sclerotic line (↑) of lamina dura (#36), residual root (#37), and horizontal impaction (#38).

**Figure 2 jfb-13-00066-f002:**
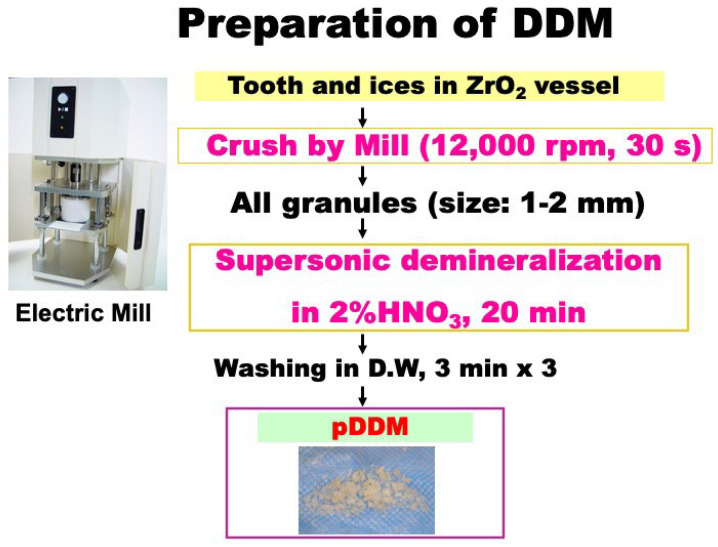
Preparation of partially demineralized dentin/cementum matrix (pDDM). ZrO_2:_ zirconium.

**Figure 3 jfb-13-00066-f003:**
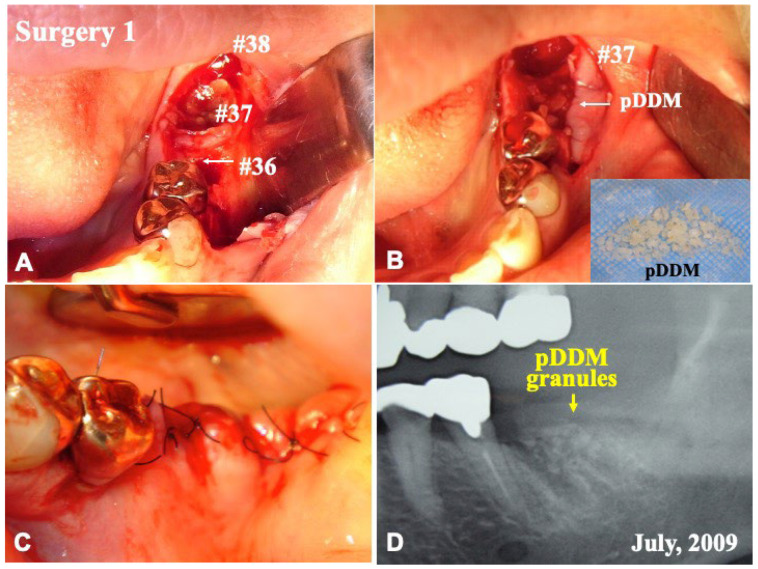
Surgery 1. (**A**) Gross view just after elevation of flap. (**B**) Immediate graft of pDDM granules into treated socket (#36). (**C**) Closed wound by sutures. (**D**) X-ray showing shadow of grafted pDDM (↓) and sockets after extraction of #37 and #38.

**Figure 4 jfb-13-00066-f004:**
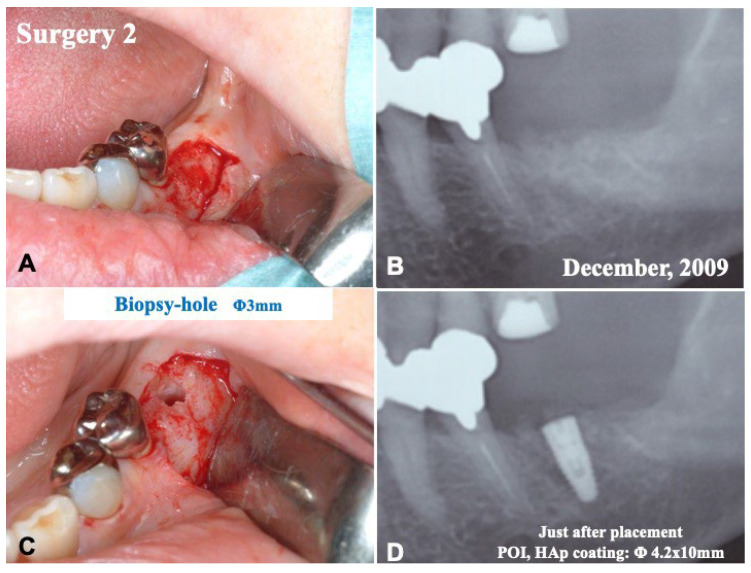
Surgery 2. (**A**) Gross view just after elevation of flap. (**B**) X-ray photo showing remodeled bone-like shadow (#36). (**C**) Biopsy (hole: 3 mm). (**D**) X-ray photo showing placement of fixture.

**Figure 5 jfb-13-00066-f005:**
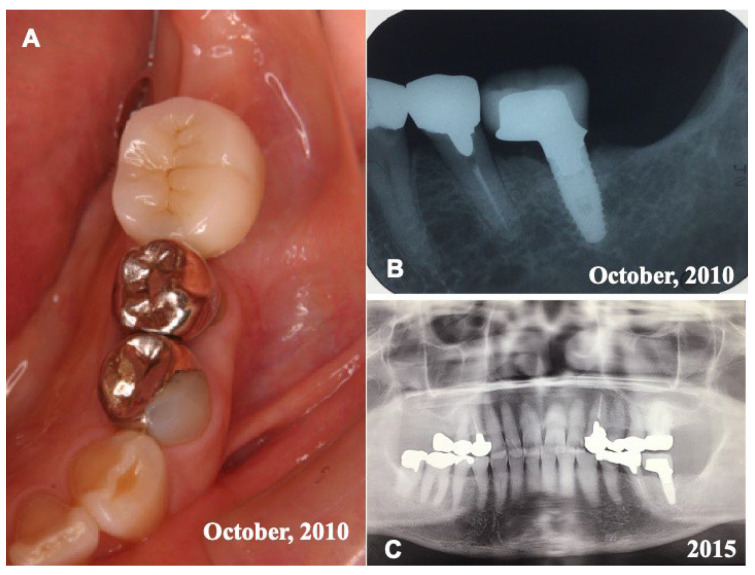
Final crown and X-ray photos. (**A**) Set of final resin crown in 2010. (**B**) X-ray photo in 2010 showing fixture with crown. (**C**) X-ray photo in 2015 showing good appearance at 6 years after placement of fixture.

**Figure 6 jfb-13-00066-f006:**
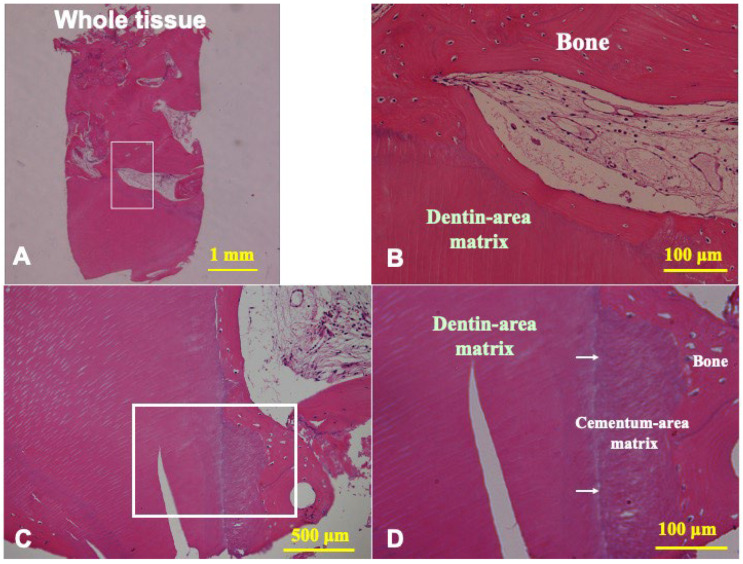
Histological photographs of HE sections of biopsy tissue at 5 months after pDDM graft. (**A**) Whole view indicating mix of newly formed bone and residues of pDDM. (**B**) Higher magnification of frame in (**A**). New bone connected with dentin-area matrix of pDDM. (**C**) pDDM including small patch of cementum. Mature bone connected with cementum- and dentin-area matrix of pDDM. (**D**) Higher magnification of frame in (**C**). Cementum-area matrix directly connected with regenerated bone. Arrows indicating boundary between cementum- and dentin-area matrices.

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
