# Peer review of "Histological Evidences of Autograft of Dentin/Cementum Granules into Unhealed Socket at 5 Months after Tooth Extraction for Implant Placement"

_jfb, 2022, doi:10.3390/jfb13020066_

Round 1

Reviewer 1 Report

  • Title is misleading; histology was performed at surgery 2, which was 5 months after surgery 1. Therefore histology evidence is for 5 months not 2. Modify the title
  • In abstract, clearly mention when the biopsy was performed.
  • In figure 6, mention when the biopsy was performed.
  • In conclusion, it is mentioned 2 months which denotes to patient not being able to heal not the point when histology was performed. Please modify and clarify. It should be 5 months

Author Response

Our paper was revised under the guidance of reviewers. Thank you very much for your valuable advice.  The term “biopsy tissue” changed into biopsy tissue at 5 months after pDDM graft, totally.

Reviewer 2 Report

Dear authors, I have no more concerns regarding this report.

Author Response

Our paper was revised under the guidance of reviewers. Thank you very much for your valuable advice.  The term “biopsy tissue” changed into biopsy tissue at 5 months after pDDM graft, totally.

This manuscript is a resubmission of an earlier submission. The following is a list of the peer review reports and author responses from that submission.

Round 1

Reviewer 1 Report

Dear Authors, 

You made a great work! However, some improvements are mandatory before acceptance. 

Reviewer 2 Report

  • English needs revision
  • Add “a case report” to the title
  • Move section 2.6 to the top and after section 2.1
  • In section 2.4, date is mentioned 2019. In the rest of the manuscript and figures, it is 2009. Why this case report which in conducted in 2009 has not been published until now?
  • Was the stability of implant measured?
  • What was the formed bone quality according to the prosthodontics guidelines?
  • When were the crowns placed? Provide radiographs on the implant with crown
  • Was there longer follow up to monitor marginal bone loss?
  • In discussion, the authors should discuss and interpret their own findings

Reviewer 3 Report

Dear authors, firstly it is important to understand that this is not a clinical article, but a case report. Therefore the submitted section should be changed. Additionally, clinical case reports aim at conveying a clinical message and despite the different types of reports, they all aim to enhance the reader’s knowledge based on novel findings or exceptional educational value regarding clinical manifestations, diagnostic approach and treatment alternatives. Thus, a clinical case report should contain practical clinical messages and educational purpose.

This procedure related with the use of demineralized dentin/cementum matrix for bone regeneration is neither new nor does it cover an extraordinary long observation period (5 months only). In contrary numerous well documented studies are published dealing with topic of demineralized dentin/cementum matrix for bone regeneration, with multiple narrative, systematic and meta-analysis reviews already available also on this same topic.  Additionally, please note the dates of the manuscript do not match with the dates of the figures.

Taking into consideration the above-mentioned aspects regarding novelty factor and educational value it is this reviewer opinion that the article should be rejected as it is.